# OPTIMIZING MIXTURE OF BLOCK ATTENTION

## ABSTRACT

Mixture of Block Attention (MoBA) (Lu et al., 2025) is a promising building block for efficiently processing long contexts in LLMs by enabling queries to sparsely attend to a small subset of key-value blocks, drastically reducing computational cost. However, the design principles governing MoBA's performance are poorly understood, and it lacks an efficient GPU implementation, hindering its practical adoption. In this paper, we first develop a statistical model to analyze MoBA's underlying mechanics. Our model reveals that performance critically depends on the router's ability to accurately distinguish relevant from irrelevant blocks based on query-key affinities. We derive a signal-to-noise ratio that formally connects architectural parameters to this retrieval accuracy. Guided by our analysis, we identify two key pathways for improvement: using smaller block sizes, and applying a short convolution on keys to cluster relevant signals, which enhances routing accuracy. While theoretically better, small block sizes are inefficient on GPUs. To bridge this gap, we introduce **FlashMoBA**, a hardware-aware CUDA kernel that enables efficient MoBA execution even with the small block sizes our theory recommends. We validate our insights by training LLMs from scratch, showing that our improved MoBA models match the performance of dense attention baselines. FlashMoBA achieves **up to 14.7×** speedup over FlashAttention-2 for small blocks, making our theoretically-grounded improvements practical.

## 1 INTRODUCTION

Large Language Models (LLMs) (Dubey et al., 2024; OpenAI, 2023) are expanding into multimodal domains like video understanding (Lin et al., 2023; Wang et al., 2024) and video generation (Kong et al., 2024), requiring the ability to handle exceptionally long contexts. This vision is bottlenecked by the self-attention mechanism (Vaswani et al., 2017), whose quadratic computational cost makes processing long sequences costly. Sparse attention (Zaheer et al., 2020; Guo et al., 2024; Xu et al., 2025) aims to solve this by focusing computation only on important regions. Among these methods, **Mixture of Block Attention (MoBA)** (Lu et al., 2025) is a promising approach where a learned router directs each query to a small subset of key-value blocks, reducing complexity to near-linear.

However, MoBA's success is hindered by two critical issues: the design principles governing its performance are poorly understood, and it lacks an efficient GPU implementation, especially for small block sizes. This raises a key question: how does the router reliably select a handful of correct blocks from thousands of candidates—a "needle-in-a-haystack" problem—and how can we make this process fast on hardware?

To answer this, we develop a statistical model of MoBA's mechanics. Our analysis reveals that retrieval accuracy is governed by a signal-to-noise ratio (SNR) that directly links architectural parameters to performance:

$$\text{SNR} \propto \sqrt{\frac{d}{B}}$$

where $d$ is the head dimension and $B$ is the block size. This insight yields two key design principles: (1) optimizing the **head-dimension-to-block-size ratio** ($d/B$), which we validate by systematically varying $B$ while holding $d$ constant, and (2) applying a **short convolution on keys** to better cluster relevant signals for the router.

While our theory advocates for small block sizes, they are notoriously inefficient on GPUs. To solve this, we introduce **FlashMoBA**, a hardware-aware CUDA kernel that makes these theoretically op-

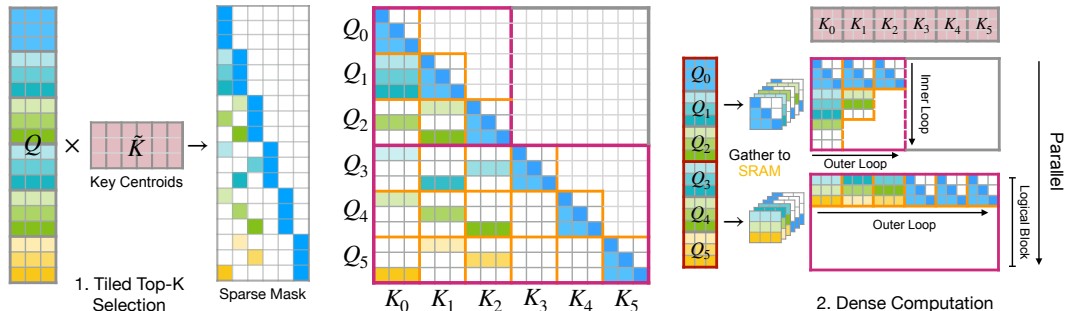

Figure 1: **FlashMoBA forward pass in two stages.** *MoBA* splits keys and values into blocks; each query scores *centroids* of key-blocks $\tilde{\mathbf{K}}$ and attends only to its top-$k$ blocks (plus causally to its own block). **1) Tiled Top-$k$ Selection:** a fused kernel streams tiles of $\mathbf{Q}$ and $\tilde{\mathbf{K}}$ to emit a sparse routing mask *without* materializing the full matrix. **2) Dense Computation:** for each selected key block, queries are *gathered* into on-chip SRAM, computed *densely* with FlashAttention-2 logic, then scattered back. This ***gather-and-densify*** strategy coalesces memory, maximizes hardware utilization and makes small-block MoBA fast on GPUs (See Section 4 for more details).

timal configurations practical. By adapting techniques from FlashAttention (Dao et al., 2022; Dao, 2023) and adding novel optimizations for block-sparsity, FlashMoBA achieves significant speedups. We validate our approach by training LLMs from scratch, demonstrating that our improved MoBA models match the performance of dense attention baselines.

Our contributions are threefold:

- **A statistical model of MoBA** that connects architectural parameters $(d, B)$ to router accuracy via a signal-to-noise ratio, providing a principled guide for design.

- **Two design principles for improving MoBA validated through controlled experiments**: optimizing the $d/B$ ratio (by varying block size $B$ while controlling for model capacity) and applying a convolution on keys to improve signal clustering.

- **FlashMoBA**, a hardware-aware CUDA kernel that makes theoretically optimal small block sizes practical, achieving **up to 14.7× speedup** on GPUs.

## 2 PRELIMINARIES

We briefly review MoBA (Lu et al., 2025). Given a sequence of $N$ key tokens, MoBA partitions them into $n = N/B$ blocks of size $B$. For each query $\mathbf{q}$, instead of attending to all $N$ keys and values, MoBA selects only the top $k$ most relevant blocks.

The block selection uses a gating mechanism. For block $i$ with keys $\mathbf{K}_i \in \mathbb{R}^{B \times d}$, the relevance score is computed as $s_i = \mathbf{q}^\top \cdot \tilde{\mathbf{k}}_i$ where $\tilde{\mathbf{k}}_i = \frac{1}{B} \sum_{\mathbf{k} \in \mathbf{K}_i} \mathbf{k}$ represents the *centroid* —i.e., mean pooling of the block $\mathbf{K}_i$. The top-$k$ blocks with highest scores are selected, and attention is computed only over their tokens:

$$\text{MoBA}(\mathbf{q}, \mathbf{K}, \mathbf{V}) = \text{softmax}(\mathbf{q}\mathbf{K}_\mathcal{S}^\top / \sqrt{d})\mathbf{V}_\mathcal{S},$$

where $\mathcal{S}$ contains all tokens from selected blocks. To preserve causality, blocks containing future tokens are masked during selection. Additionally, each query always attends to its current block with causal masking.

This reduces complexity from $O(N^2)$ to $O(N \cdot kB)$ when $k \ll n$. Our experiments systematically explore configurations with $B \in \{512, 256, 128\}$ and corresponding $k \in \{2, 4, 8\}$ to maintain constant sparsity of 7/8 when $N = 8192$. To encourage within-block clustering, we optionally apply a short depthwise causal convolution on keys (Yang et al., 2024) (kernel size 3 or 5); details are provided in Appendix C.

## 3   A STATISTICAL MODEL OF MOBA

For MoBA to be effective, its router must select the correct key-value block for a given query. This is challenging because the router scores a block using its centroid (the average of all its keys), a process that risks drowning out the signal from a single relevant token. To understand how MoBA succeeds, we developed a statistical model of its block selection mechanism.

### 3.1   MODELING THE BLOCK SELECTION CHALLENGE

We model the router's task by treating the dot products between a query $\mathbf{q}$ and the keys as random variables. We assume that for a given query, "signal" keys ($\mathbf{k}^*$) it is seeking have a higher expected dot product than irrelevant "noise" keys ($\mathbf{k}$). Let $\mu_{\text{signal}} = \mathbb{E}[\mathbf{q}^\top \mathbf{k}^*]$ and $\mu_{\text{noise}} = \mathbb{E}[\mathbf{q}^\top \mathbf{k}]$. The fundamental separation between signal and noise is the difference $\Delta\mu = \mu_{\text{signal}} - \mu_{\text{noise}}$. For a router to function, this separation must be positive ($\Delta\mu > 0$).

The router scores each block $j$ by computing the dot product with the block's centroid, $s_j = \mathbf{q}^\top \tilde{\mathbf{k}}_j$, where $\tilde{\mathbf{k}}_j = \frac{1}{B}\sum_{\mathbf{k}\in\text{block}_j} \mathbf{k}$. The key question is whether the score of the signal-containing block ($s_{j^*}$) will reliably be higher than the score of any noise block ($s_j$).

### 3.2   SIGNAL-TO-NOISE RATIO (SNR) ANALYSIS

To quantify when block selection succeeds, we analyze the **signal-to-noise ratio (SNR)** of the router's scores. We consider the difference in scores, $D = s_{j^*} - s_j$, between the signal block $j^*$ and a pure noise block $j$. The expected value of this difference represents the "signal," while its standard deviation represents the "noise."

Through statistical analysis (see Appendix B), we find:

$$\mathbb{E}[D] = \frac{\Delta\mu_{\text{eff}}}{B} \tag{1}$$

$$\text{Var}(D) \approx \frac{2}{dB} \quad \text{(for normalized vectors)} \tag{2}$$

Here, $\Delta\mu_{\text{eff}}$ is the **effective signal separation**. If $m$ related signal tokens are clustered within the target block, this separation is amplified: $\Delta\mu_{\text{eff}} = \Delta\mu + (m-1)(\mu_{\text{cluster}} - \mu_{\text{noise}})$, where $\mu_{\text{cluster}}$ is the average affinity between the query and other clustered signal tokens.

The SNR is the ratio of the expected outcome to its standard deviation, which is our central finding:

$$\text{SNR} = \frac{\mathbb{E}[D]}{\sqrt{\text{Var}(D)}} = \Delta\mu_{\text{eff}}\sqrt{\frac{d}{2B}} \tag{3}$$

The probability of a retrieval failure—a noise block outranking the signal block—decreases exponentially as the SNR increases: $p_{\text{fail}} = \Phi(-\text{SNR})$, where $\Phi$ is the standard normal CDF. For reliable top-$k$ retrieval in a long context with thousands of blocks, a high SNR is essential.

### 3.3   ARCHITECTURAL INSIGHTS FROM THE SNR MODEL

The SNR formula provides two clear and actionable principles for designing effective MoBA architectures:

**1. The $d/B$ ratio is the key.** The SNR is proportional to $\sqrt{d/B}$. This ratio emerges as the most critical factor governing the router's retrieval capability. This insight suggests two avenues for improvement: increasing the head dimension $d$ or decreasing the block size $B$. However, the head dimension $d$ is a *confounding variable*; increasing it not only improves the theoretical SNR but also increases the model's parameters and computational cost (FLOPs), adding capacity that makes a controlled comparison difficult. Therefore, to isolate and empirically validate the $d/B$ ratio's impact on retrieval accuracy alone, our study fixes $d$ (controlling for model capacity) and systematically varies $B$. Halving the block size improves the SNR by a factor of $\sqrt{2}$, making it easier for the router to find the signal.

**2. Within-block clustering is a performance multiplier.** When semantically related tokens are grouped together in a block—a behavior encouraged by token-level key convolution (Yang et al., 2024) during training—the effective signal $\Delta\mu_{\text{eff}}$ increases via larger $m$ and $\mu_{\text{cluster}}$, significantly boosting the SNR.

These principles form the theoretical foundation for our architectural improvements, which we validate systematically in Section 5.

## 4 FLASHMOBA: AN OPTIMIZED KERNEL FOR SMALL-BLOCK MOBA

Our theoretical model shows that **smaller block sizes** yield significant quality gains, but a naive GPU implementation is inefficient. The original MoBA implementation released by Lu et al. (2025), when configured with small blocks, suffers from performance bottlenecks that negate the computational savings from sparsity, resulting in slower execution than dense attention. We introduce **FlashMoBA**, an hardware-aware CUDA kernel designed to make small-block MoBA practical and fast.

### 4.1 PERFORMANCE CHALLENGES WITH SMALL BLOCKS

Small block sizes introduce several critical performance challenges that must be addressed for practical deployment. First, **inefficient memory access** occurs when gathering sparse, non-contiguous key-value blocks for each query, leading to uncoalesced memory reads from HBM. Second, **top-k and gating overhead** becomes problematic as smaller block sizes $B$ increase the number of blocks ($n = N/B$) that the router must score. The original implementation materializes a large $N \times n$ score matrix, incurring substantial memory overheads. Finally, **low GPU occupancy** results from the reduced work per block and the overhead of launching many independent kernels, leading to poor parallelism and low hardware utilization.

### 4.2 FLASHMOBA KERNEL DESIGN

To overcome these challenges, FlashMoBA employs three fused kernels that minimize HBM round-trips and aligns computation with the GPU architecture, as depicted in Figure 1.

**1. Tiled Top-K Selection** The top-k selection process is a primary bottleneck in the original MoBA implementation Lu et al. (2025), which materializes a full score matrix and processes batched sequences serially. We replace this with **Flash TopK** (Step 1 in Figure 1), a highly optimized three-stage pipeline of fused kernels. First, a Triton kernel computes key-block centroids, producing a much smaller matrix $\tilde{\mathbf{K}}$. Second, a tiled kernel inspired by FlashAttention-2 finds the top-k key-blocks for each query by computing scores between $\mathbf{Q}$ and $\tilde{\mathbf{K}}$ *without ever materializing the full score matrix to HBM*, as summarized in Algorithm 3. Finally, an efficient epilogue reformats the query-centric indices into a key-block-centric varlen layout for the main attention pass. This entire pipeline is fully parallelized across batches and heads, eliminating the original performance bottleneck. The detailed breakdown of all three stages is provided in Appendix D.1.

**2. Forward Pass with Gather-and-Densify** To handle MoBA's irregular sparsity, our forward kernel uses a "gather-and-densify" strategy based on a two-level blocking mechanism, detailed in Algorithm 1. We distinguish between two types of blocks:

- **Logical Blocks:** Large, contiguous blocks of queries ($\mathbf{Q_i}$) and keys ($\mathbf{K_j}$) that the kernel iterates over in its outer loops. A logical key block matches a MoBA key block.

- **Physical Blocks:** Smaller tiles (e.g., $64 \times 64$ or $128 \times 128$) loaded into SRAM for matrix multiplication. Their optimal size depends on GPU architecture and head dimension.

The kernel assigns a logical query block $\mathbf{Q_i}$ to each thread block, iterating through all logical key blocks $\mathbf{K_j}$. For each pair, it uses varlen indices to find relevant queries. This subset is batched into dense physical blocks: a physical block of queries is gathered from HBM into a dense SRAM buffer for computation. This two-level approach is key, as caching the queries in SRAM allows reuse across all physical tiles of the logical key block, amortizing costly irregular memory access with efficient dense GEMMs.

---

**Algorithm 1** FlashMoBA Forward Pass

---

**Require:** Matrices $\mathbf{Q}, \mathbf{K}, \mathbf{V} \in \mathbb{R}^{N \times d}$ in HBM. Varlen indices $A$, Offset, $C$. Logical block sizes $B_q, B$. Physical tile sizes $B_r, B_c$.

1: Divide $\mathbf{Q}$ into $T_q = \lceil \frac{N}{B_q} \rceil$ logical blocks $\mathbf{Q}_i$ of size $B_q \times d$.

2: Divide $\mathbf{K}, \mathbf{V}$ into $T_k = \lceil \frac{N}{B} \rceil$ logical blocks $\mathbf{K}_j, \mathbf{V}_j$ of size $B \times d$.

3: Initialize output matrix $\mathbf{O} \in \mathbb{R}^{N \times d}$, temporary output matrix $\mathbf{O_{tmp}} \in \mathbb{R}^{N \times d}$ and logsumexp vector $L \in \mathbb{R}^N$ in HBM.

4: **for** $0 \leq i < T_q$ **do in parallel**

5:     **for** $0 \leq j < T_k$ **do**

6:         Let Indices$_j$ be the list of $N_j$ query indices from $\mathbf{Q}_i$ attending to $\mathbf{K}_j$.

7:         Define tile counts $T_c = \lceil B/B_c \rceil, T_r^{(j)} = \lceil N_j/B_r \rceil$.

8:         Divide $\mathbf{K}_j, \mathbf{V}_j$ into physical tiles $\mathbf{K}_{j,k}, \mathbf{V}_{j,k}$.

9:         **for** $0 \leq r < T_r^{(j)}$ **do**

10:             Gather-load the $r$-th tile of queries from $\mathbf{Q}$ into SRAM based on Indices$_j$.

11:             Gather-load the corresponding outputs from $\mathbf{O_{tmp}}$ into on-chip accumulators.

12:             **for** $0 \leq k < T_c$ **do**

13:                 Load $\mathbf{K}_{j,k}, \mathbf{V}_{j,k}$ from HBM to SRAM.

14:                 On-chip, compute scores $\mathbf{S}_{rk} = \mathbf{Q}_r^{(j)}(\mathbf{K}_{j,k})^\top$.

15:                 Update on-chip softmax stats and partial outputs in accumulators.

16:             **end for**

17:             Scatter-store the updated partial outputs from accumulators back to $\mathbf{O_{tmp}}$.

18:         **end for**

19:     **end for**

20:     Load data from $\mathbf{O_{tmp}}$, convert to output dtype, and write back to $\mathbf{O}_i$.

21: **end for**

---

Table 1: Performance comparison on language modeling and zero-shot common-sense reasoning for 340M models trained on 100B tokens. MoBA-128 + kconv5 achieves the best average performance.

| Model | Wiki ppl ↓ | OBQA acc ↑ | PIQA acc ↑ | Hella. acc ↑ | Lamb. acc ↑ | ARC-c acc ↑ | TQA acc ↑ | ARC-e acc ↑ | Wino. acc ↑ | Avg. acc ↑ |
|---|---|---|---|---|---|---|---|---|---|---|
| Dense | 19.6 | 20.8 | 69.7 | 48.5 | 39.8 | 30.5 | 27.1 | 63.8 | 53.5 | 44.2 |
| MoBA-512 | 20.9 | 22.0 | 68.7 | 48.2 | 39.7 | 31.1 | 29.7 | 64.3 | 53.4 | 44.6 |
| MoBA-256 | 20.3 | 22.8 | 68.8 | **48.5** | 39.2 | 30.9 | 27.8 | 63.4 | 55.1 | 44.6 |
| MoBA-128 | 19.7 | 21.8 | 69.0 | 48.3 | 40.9 | 31.7 | 28.2 | **65.5** | 55.4 | 45.1 |
| + kconv3 | **19.3** | **25.6** | **69.7** | 48.3 | **41.7** | **32.6** | 26.8 | 64.7 | 55.1 | 45.6 |
| + kconv5 | 19.5 | 23.2 | 68.9 | 48.4 | 40.0 | 31.7 | **36.1** | 64.8 | **56.3** | **46.2** |

**3. Backward Pass with Recomputation** Our backward pass leverages the memory-efficient design of FlashAttention-2 and is implemented as a sequence of three kernels (Algorithm 5). The primary kernel parallelizes computation across the key dimension, with each thread block processing one key-block. To handle sparsity, it mirrors the forward pass's "gather-and-densify" strategy, using varlen indices to gather subsets of queries and output gradients into on-chip tiles. Following the FlashAttention-2 methodology, we recompute attention scores during the backward pass to avoid storing the full attention matrix in memory. While key and value gradients are written directly to HBM, the partial query gradients ($\mathbf{dQ}$) require accumulation across multiple key-blocks, which is handled efficiently and safely using atomic additions to a high-precision global buffer. This design ensures that the backward pass maintains linear complexity in sequence length, a critical improvement over the quadratic complexity of standard attention. As the backward pass typically constitutes the main performance bottleneck in optimized attention implementations (often 2-3× slower than the forward pass Dao (2023)), the efficiency of our backward kernel is crucial for enabling practical training on long sequences.

Table 2: Performance comparison on language modeling and zero-shot common-sense reasoning for 1B models trained on 100B tokens. MoBA-128 + kconv3 achieves the best average performance.

| Model | Wiki ppl ↓ | OBQA acc ↑ | PIQA acc ↑ | Hella. acc ↑ | Lamb. acc ↑ | ARC-c acc ↑ | TQA acc ↑ | ARC-e acc ↑ | Wino. acc ↑ | Avg. acc ↑ |
|---|---|---|---|---|---|---|---|---|---|---|
| Dense | 14.7 | 26.2 | 72.6 | 59.4 | 47.1 | 38.4 | 33.0 | 71.3 | 59.0 | 50.9 |
| MoBA-128 | 15.1 | 26.2 | **73.6** | 59.7 | 48.0 | **41.4** | 30.2 | 73.0 | **61.5** | 51.7 |
| + kconv3 | **14.5** | 25.4 | 72.6 | **60.1** | 48.1 | 40.4 | **43.5** | 72.5 | 58.7 | **52.7** |
| + kconv5 | 14.7 | **27.8** | 73.1 | 59.7 | **49.0** | 39.9 | 30.5 | **73.3** | 59.6 | 51.6 |

Table 3: Zero-shot performance on RULER S-NIAH tasks for 340M models trained on 100B tokens. Models trained on 8K contexts are evaluated on sequences up to 64K without fine-tuning. The reported scores are accuracy percentages from a 1000-sample test set.

| Model | S-NIAH-1 | | | | | S-NIAH-2 | | | | | S-NIAH-3 | | | | | Avg. |
|---|---|---|---|---|---|---|---|---|---|---|---|---|---|---|---|---|
| | 4K | 8K | 16K | 32K | 64K | 4K | 8K | 16K | 32K | 64K | 4K | 8K | 16K | 32K | 64K | |
| Dense | **100** | **100** | 79 | 0 | 0 | **100** | 99 | 5 | 0 | 0 | **78** | **69** | 0 | 0 | 0 | 42.0 |
| MoBA-512 | 90 | 86 | 72 | 59 | 35 | 81 | 41 | 14 | 4 | **1** | 63 | 30 | **7** | **1** | 0 | 38.8 |
| MoBA-256 | **100** | **100** | **100** | 99 | 94 | 96 | 64 | 22 | **5** | 0 | 37 | 14 | **5** | 0 | 0 | 49.1 |
| MoBA-128 | **100** | **100** | **100** | **100** | 85 | 99 | 92 | **65** | **17** | 1 | 52 | 25 | **5** | 0 | 0 | 56.0 |
| + kconv3 | **100** | **100** | **100** | 99 | 96 | 99 | 94 | 58 | 5 | 0 | 57 | 22 | 4 | 0 | 0 | 55.5 |
| + kconv5 | **100** | **100** | **100** | **100** | **100** | **100** | 99 | **71** | 3 | 0 | **95** | **67** | **22** | **1** | 0 | **63.9** |

Table 4: Zero-shot performance on RULER S-NIAH tasks for 1B models. Models trained on 8K contexts are evaluated on sequences up to 64K without fine-tuning. The reported scores are accuracy percentages from a 1000-sample test set.

| Model | S-NIAH-1 | | | | | S-NIAH-2 | | | | | S-NIAH-3 | | | | | Avg. |
|---|---|---|---|---|---|---|---|---|---|---|---|---|---|---|---|---|
| | 4K | 8K | 16K | 32K | 64K | 4K | 8K | 16K | 32K | 64K | 4K | 8K | 16K | 32K | 64K | |
| Dense | **100** | **100** | **100** | 62 | 0 | **100** | 99 | 91 | 0 | 0 | 94 | 92 | **81** | 0 | 0 | 61.3 |
| MoBA-128 | **100** | **100** | **100** | 99 | 74 | **100** | 98 | 83 | 16 | 1 | 81 | 67 | 29 | 2 | 0 | 63.3 |
| + kconv3 | **100** | **100** | **100** | **100** | 66 | **100** | **100** | 95 | **37** | 2 | 89 | 78 | 45 | **10** | **1** | **68.2** |
| + kconv5 | **100** | **100** | **100** | **100** | **100** | **100** | 96 | 76 | 17 | 0 | **98** | 84 | 44 | 7 | 0 | 68.1 |

## 5 EXPERIMENTS

### 5.1 EXPERIMENTAL SETUP

We validate our design principles of MoBA through controlled experiments on models pre-trained from scratch.

**Model Architecture.** All models use a hybrid 24-layer architecture: odd layers use sliding window attention (window size 256) with RoPE, while even layers use either dense attention (baseline) or MoBA variants without positional encoding. This design, inspired by Command-A (Team Cohere, 2025) and SWAN-GPT (Puvvada et al., 2025), isolates MoBA's contribution while maintaining local dependencies. We train two model families: **340M** (hidden size 1024, 16 heads, intermediate size 2816) and **1B** (hidden size 2048, 32 heads, intermediate size 8192). Both model families use a fixed head dimension $d = 64$. While our SNR model predicts that increasing $d$ is beneficial, this change is confounded with an increase in model parameters and FLOPs. To conduct a controlled experiment that isolates the effect of the router's retrieval mechanism (as governed by the $d/B$ ratio) from changes in model capacity, we fix $d$ and focus our empirical validation on the effect of block size $B$ and key convolution. Both use Llama-2 tokenizer (32K vocabulary) and 8K training context.

Table 5: Performance on real-world LongBench tasks for 340M models trained on 100B tokens. MoBA-128 + kconv3 achieves the best average scores.

| Model | Single-Doc QA | | Multi-Doc QA | | | Summarization | | | Few-shot | | Code | | Avg. |
|---|---|---|---|---|---|---|---|---|---|---|---|---|---|
| | Qasper | MField | HotpQA | 2Wiki | MuSiQue | GovR | QMSum | MNews | Trivia | SAMSum | LCode | RepoB | |
| Dense | 7.6 | **17.3** | 4.0 | 9.1 | 2.3 | 12.1 | 15.2 | 12.5 | 8.3 | 11.8 | 19.1 | 16.9 | 11.3 |
| MoBA-512 | 6.5 | 14.4 | 5.8 | **10.6** | 3.7 | 13.7 | 16.8 | 13.2 | 10.8 | 15.0 | 16.7 | 21.1 | 12.4 |
| MoBA-256 | 7.3 | 15.3 | 6.1 | 10.3 | 3.6 | 13.8 | 17.6 | 12.9 | 11.2 | 14.9 | **22.3** | **22.5** | 13.2 |
| MoBA-128 | 6.8 | 15.2 | 5.6 | 10.2 | 3.7 | 13.0 | 16.1 | 11.2 | 11.3 | 16.8 | 20.8 | 19.1 | 12.5 |
| + kconv3 | **8.3** | 14.4 | **6.5** | 9.5 | 3.9 | 14.3 | **18.3** | **19.4** | **11.6** | 16.3 | 21.3 | 20.4 | **13.7** |
| + kconv5 | 7.4 | 14.3 | 6.3 | 10.1 | **4.0** | **17.8** | 17.4 | 18.4 | 10.5 | **17.8** | 15.6 | 17.6 | 13.1 |

Table 6: Performance on real-world LongBench tasks for 1B models trained on 100B tokens.

| Model | Single-Doc QA | | Multi-Doc QA | | | Summarization | | | Few-shot | | Code | | Avg. |
|---|---|---|---|---|---|---|---|---|---|---|---|---|---|
| | Qasper | MField | HotpQA | 2Wiki | MuSiQue | GovR | QMSum | MNews | Trivia | SAMSum | LCode | RepoB | |
| Dense | 9.2 | 19.0 | 6.3 | **10.7** | 4.4 | **22.7** | 17.1 | **18.7** | 12.8 | 16.7 | 20.3 | 18.1 | 14.6 |
| MoBA-128 | 8.9 | **20.2** | 7.4 | 9.2 | 4.1 | 16.4 | 18.1 | 17.4 | 14.2 | **19.0** | 18.1 | 21.1 | 14.5 |
| + kconv3 | 8.2 | 17.7 | 7.4 | 8.9 | **4.5** | 22.3 | 18.4 | 12.5 | 13.7 | 18.1 | 21.7 | 21.5 | 14.6 |
| + kconv5 | 8.7 | 18.3 | 7.0 | 9.5 | 4.2 | 19.5 | 18.5 | 18.4 | 13.2 | 18.6 | **21.8** | 23.4 | **15.1** |

**Configurations.** For 340M models, we systematically vary block sizes while maintaining 7/8 sparsity: **MoBA-512** ($B = 512$, $k = 2$), **MoBA-256** ($B = 256$, $k = 4$), and **MoBA-128** ($B = 128$, $k = 8$). For 1B models, we focus on the MoBA-128 configuration. Our theory predicts smaller $B$ improves SNR and retrieval accuracy.

**Training and Evaluation.** All models are pre-trained on FineWeb-Edu (Penedo et al., 2024) using AdamW optimizer with $\beta_1 = 0.9$, $\beta_2 = 0.95$, weight decay 0.1, and cosine learning rate schedule. We use peak LR $6 \times 10^{-4}$, batch size 500K tokens. All models are trained on 100B tokens. All models use gradient clipping at 1.0, and mixed precision training with bfloat16. We evaluate on: **(1) Language Modeling:** WikiText2 perplexity (Merity et al., 2017) and 8 zero-shot tasks: OpenBookQA (Mihaylov et al., 2018), PIQA (Bisk et al., 2020), HellaSwag (Zellers et al., 2019), WinoGrande (Sakaguchi et al., 2020), ARC-e/c (Clark et al., 2018), TruthfulQA (Lin et al., 2021), and LAMBADA (Paperno et al., 2016). **(2) Long-Context Retrieval:** S-NIAH-1/2/3 from RULER (Hsieh et al., 2024) at 4K–64K lengths. **(3) LongBench:** 12 tasks (Bai et al., 2023): single-document QA (Qasper, MField), multi-document QA (HotpotQA, 2WikiMQA, MuSiQue), summarization (GovReport, QMSum, MultiNews), few-shot (TriviaQA, SAMSum), and code (LCC, RepoBench-P).

**Implementation.** Models are implemented using PyTorch with FlashAttention-2 (Dao, 2023) for efficient attention computation. For MoBA, we compare our custom CUDA kernels (detailed in Section 4) against the original implementation released by (Lu et al., 2025). Our kernels build upon FlashAttention's tiling strategy while introducing novel optimizations specifically for block-sparse patterns with small blocks. All experiments use 8×H100 80GB GPUs with gradient checkpointing and fully-sharded data parallelism for memory efficiency.

## 5.2 QUALITY RESULTS

We evaluate MoBA's quality across language modeling, long-context retrieval, and real-world tasks. Our experiments confirm that the theoretical improvements translate to consistent gains across diverse benchmarks.

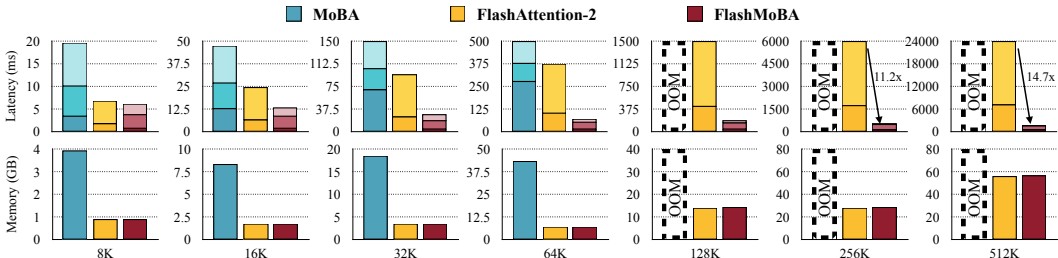

Figure 3: **Latency & memory vs. length** (bsz=2, $B$=128, $k$=8) for MoBA (original), FlashAttention-2, and **FlashMoBA**. Top: end-to-end latency; bottom: peak memory. MoBA/FlashMoBA bars are decomposed (top→bottom) into *backward*, *forward*, and *Top-k* overheads. MoBA is dominated by non-attention overheads and hits **OOM at 128K**. By fusing tiled Top-$k$ with a gather-and-densify kernel, **FlashMoBA** makes overhead negligible, cuts memory, and is **up to 14.7× faster** than FlashAttention at long sequence lengths.

**Block Size Impact.** Figure 2 shows block size effects on WikiText perplexity and RULER accuracy for 340M models. As predicted by SNR $\propto 1/\sqrt{B}$, reducing block size from 512 to 128 improves perplexity from 20.9 to 19.7 and RULER from 38.8% to 56.0%. Smaller blocks help the router identify relevant content more precisely.

The trend holds across all benchmarks and scales. For 340M models, reducing block size 4× from 512 to 128 improves language modeling accuracy from 44.6% to 45.6% (Table 1), RULER from 38.8% to 63.9% (Table 3), and LongBench from 13.2 to 15.3 (Table 5). Small blocks are necessary for MoBA to match dense attention.

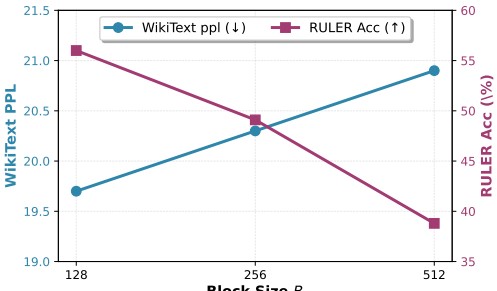

Figure 2: Smaller block sizes improve WikiText perplexity and RULER accuracy (340M, $d = 64$, 100B tokens). Reducing $B$ from 512 to 128 lowers ppl by 1.2 and increases RULER by 17.2%.

**Key Convolution Benefits.** Key convolution improves performance with task-specific preferences. For 340M models, kconv3 increases commonsense reasoning from 45.1% to 45.6% (Table 1), while kconv5 achieves 100% retrieval at 64K vs 85% without (Table 3). On LongBench, kconv3 reaches 15.3% (Table 5). At 1B scale, kconv3 improves LM accuracy to 52.7% (Table 2) and RULER to 68.2% (Table 4). These gains confirm convolution amplifies $\Delta\mu_{\text{eff}}$ by clustering related tokens.

**Sparse Matching Dense.** Across multiple benchmarks and scales, MoBA matches or even surpasses dense attention: for example, at the 340M scale, MoBA-128 + kconv3 achieves 15.3% accuracy on LongBench versus dense's 12.9% (Table 5); on RULER, dense attention fails completely at long contexts (0% at 32K tokens), while MoBA-128 + kconv5 achieves 100% at 64K (Table 3); and at 1B scale, MoBA achieves 52.7% LM accuracy compared to dense's 50.9% (Table 2), 69.6% vs 61.3% on RULER (Table 4), and 15.1% vs 14.7% on LongBench (Table 6). These results demonstrate the benefit of reducing attention dilution: dense softmax spreads probability mass thinly across all tokens as sequence length grows, making it harder to focus on relevant information, while MoBA's sparse routing concentrates attention on a small number of targeted blocks, mitigating dilution and allowing the model to more effectively select and aggregate information. This mechanism explains how MoBA is able to consistently match or outperform dense attention across diverse settings.

## 5.3 Efficiency Results

While theory favors small blocks for quality, they were previously impractical due to poor GPU utilization. FlashMoBA makes these configurations efficient.

**End-to-End Performance.** Figure 3 compares latency and memory consumption across sequence lengths from 8K to 512K tokens. FlashMoBA reduces both latency and memory significantly. At $N$=64K with $B$=128, FlashMoBA is 7.4× faster than original MoBA and uses 6.1× less memory. Original MoBA hits OOM at 128K, while FlashMoBA scales to 512K. The advantage grows with longer sequences and smaller blocks because FlashMoBA eliminates global reindexing overhead, achieving up to 14.7× speedup over FlashAttention-2 at long sequences.

**Breakdown Analysis.** To understand where FlashMoBA's speedup comes from, Figure 4 shows the forward-pass timing breakdown at $N$=64K. Original MoBA (Lu et al., 2025) uses five stages: (1) compute centroids and top-$k$, (2) global reindexing, (3) attention on routed indices, (4) local causal attention, (5) merge results. Stages (1), (2), and (5) dominate runtime, accounting for over 70% of execution time due to materialization and reindexing overhead. FlashMoBA uses two fused kernels: (i) centroid/gating/top-$k$ without materialization; (ii) gather-and-densify for attention with high occupancy. This reduces forward-pass runtime to 49 ms at $N$=64K compared to FlashAttention-2's 99 ms.

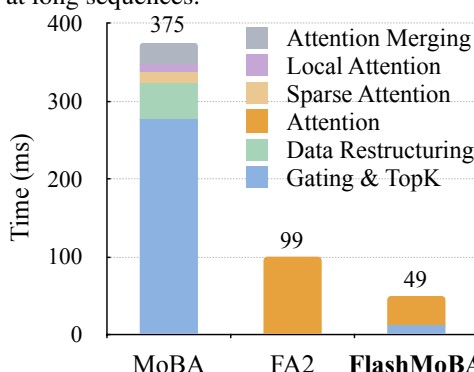

Figure 4: **Forward-pass timing breakdown** ($N$=64K, $B$=128, $k$=8). MoBA (original) is bottlenecked by routing overheads, while FlashMoBA's fused kernels cut total time to 49 ms, outperforming FlashAttention-2.

## 6 RELATED WORK

**Efficient Attention Mechanisms** The quadratic complexity of attention has driven research into efficient alternatives. Fixed-pattern methods include Sparse Transformer (Child et al., 2019), Longformer (Beltagy et al., 2020), and BigBird (Zaheer et al., 2020). Reformer (Kitaev et al., 2020) uses LSH, while Linformer (Wang et al., 2020) uses projection. Learnable approaches include Routing Transformer (Roy et al., 2021) and Performer (Choromanski et al., 2021). FlashAttention (Dao et al., 2022; Dao, 2023) improves implementation via IO-aware algorithms but doesn't reduce complexity.

**Block Sparse Attention** Block-based methods reduce complexity from $O(N^2)$ to $O(N \cdot B \cdot k)$. Blockwise Transformer (Qiu et al., 2020) pioneered this approach. Recent methods like Block Sparse Attention (Guo et al., 2024) and XAttention (Xu et al., 2025) refine block selection. Native sparse methods like MoBA (Lu et al., 2025) and Native Sparse Attention (Yuan et al., 2025) train from scratch with sparsity. Post-training methods (Zhang et al., 2023; Xiao et al., 2023; Tang et al., 2024; Jiang et al., 2024; Lai, 2025) prune existing models. Our work provides theoretical analysis of why MoBA works via a signal-to-noise ratio that guides design.

**Implementation** Sparse patterns are challenging to implement efficiently due to irregular memory access. While Triton (Tillet et al., 2019) simplifies kernel development, peak performance requires careful optimization (Hong et al., 2023; Kwon et al., 2023; Liu et al., 2023; Ye et al., 2025). FlashMoBA enables practical deployment of small-block configurations.

## 7 CONCLUSION

We presented a statistical framework for understanding Mixture of Block Attention. Our analysis reveals the mechanism behind its success: a signal-to-noise ratio SNR $= \Delta\mu_{\text{eff}} \sqrt{\frac{d}{2B}}$ that governs block selection accuracy. This insight yields key design principles validated through controlled experiments: optimizing the $d/B$ ratio (which we validate by varying $B$ while controlling for model capacity) and using key convolution to enhance signal clustering. Optimized MoBA matches or exceeds dense attention on real-world tasks while using 12.5% of the computation.

To make small blocks practical, we developed FlashMoBA, achieving up to 14.7× speedup over FlashAttention-2. This enables deployment of theoretically optimal configurations that were previously impractical. For applications requiring million-token contexts, our work shows that theoretical analysis combined with efficient implementation can scale beyond dense attention's limits.

## REPRODUCIBILITY STATEMENT.

To ensure reproducibility of our work, we provide comprehensive details throughout the paper and appendices. Our theoretical contributions include complete proofs: the SNR formula derivation is presented in Section 3 with full mathematical details in Appendix B. All model architectures and hyperparameters are specified in Section 5, including the hybrid SWA-MoBA design, hidden dimensions, and head configurations. Training details are fully described: we use publicly available FineWeb-Edu dataset (Penedo et al., 2024) with all experiments using 100B tokens. Our evaluation uses standard benchmarks with exact task specifications: RULER (Hsieh et al., 2024) and LongBench (Bai et al., 2023). The FlashMoBA kernel implementation is detailed in Section 4 with algorithmic pseudocode, and complete implementation details appear in Appendix B. We will release our code including model implementations, FlashMoBA CUDA kernels, and evaluation scripts upon publication. All experiments were conducted on 8× H100 80GB GPUs with PyTorch and FlashAttention-2 (Dao, 2023), enabling exact reproduction of our results.

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

## A  LLM USAGE STATEMENT

We acknowledge the use of Large Language Models (specifically Claude and GPT-5) in the preparation of this manuscript. The LLMs were used exclusively as writing assistants to:

- Polish and refine the language for clarity and conciseness
- Improve grammar and sentence structure
- Suggest alternative phrasings for technical descriptions
- Help organize and structure sections for better flow

## B  DETAILED DERIVATION OF SIGNAL-TO-NOISE RATIO

We provide the complete mathematical derivation of the SNR formula for MoBA's block selection mechanism.

### B.1  PROBLEM SETUP

Consider a query $\mathbf{q}_i$ seeking information from a specific key $\mathbf{k}^*$ (the signal) among $N$ keys. Let $j^*$ denote the block containing $\mathbf{k}^*$. MoBA computes similarity scores $s_j = \mathbf{q}_i^\top \tilde{\mathbf{k}}_j$ where $\tilde{\mathbf{k}}_j = \frac{1}{B} \sum_{\mathbf{k} \in \mathbf{K}_j} \mathbf{k}$ is the centroid of block $j$.

For the signal block $j^*$:

$$s_{j^*} = \frac{1}{B} \left( \mathbf{q}_i^\top \mathbf{k}^* + \sum_{\mathbf{k} \in \mathbf{K}_{j^*} \setminus \{\mathbf{k}^*\}} \mathbf{q}_i^\top \mathbf{k} \right)$$

We define successful retrieval as $\mathrm{rank}(s_{j^*}) \leq k$.

### B.2  STATISTICAL MODELING

We model the expected dot products as:

$$\mathbb{E}[\mathbf{q}_i^\top \mathbf{k}^*] = \mu_{\text{signal}} \tag{4}$$

$$\mathbb{E}[\mathbf{q}_i^\top \mathbf{k}] = \mu_{\text{noise}} \quad \text{for } \mathbf{k} \neq \mathbf{k}^* \tag{5}$$

Define the similarity gap $\Delta\mu = \mu_{\text{signal}} - \mu_{\text{noise}} > 0$.

To account for semantic clustering, let $m$ denote the number of keys in the signal block with elevated similarity $\mu_{\text{cluster}}$ where $\mu_{\text{noise}} < \mu_{\text{cluster}} \leq \mu_{\text{signal}}$. The expected scores become:

$$\mathbb{E}[s_{j^*}] = \frac{1}{B}[\mu_{\text{signal}} + (m-1)\mu_{\text{cluster}} + (B-m)\mu_{\text{noise}}] \tag{6}$$

$$\mathbb{E}[s_j] = \mu_{\text{noise}} \quad \text{for } j \neq j^* \tag{7}$$

The expected advantage of the signal block is:

$$\mathbb{E}[s_{j^*}] - \mathbb{E}[s_j] = \frac{\Delta\mu + (m-1)(\mu_{\text{cluster}} - \mu_{\text{noise}})}{B} = \frac{\Delta\mu_{\text{eff}}}{B}$$

where $\Delta\mu_{\text{eff}} = \Delta\mu + (m-1)(\mu_{\text{cluster}} - \mu_{\text{noise}})$ is the effective similarity gap.

### B.3  VARIANCE ANALYSIS

Assuming independent dot products with variance $\sigma^2 \approx 1/d$ for normalized vectors in high dimensions, the score difference $D = s_{j^*} - s_j$ between signal and noise blocks satisfies:

$$\mathbb{E}[D] = \frac{\Delta\mu_{\text{eff}}}{B} \tag{8}$$

$$\mathrm{Var}(D) = \mathrm{Var}(s_{j^*}) + \mathrm{Var}(s_j) = \frac{\sigma^2}{B} + \frac{\sigma^2}{B} = \frac{2\sigma^2}{B} \tag{9}$$

### B.4 RETRIEVAL PROBABILITY

Under the Central Limit Theorem for large $B$, $D \sim \mathcal{N}(\Delta\mu_{\text{eff}}/B, 2\sigma^2/B)$. The probability that a noise block outranks the signal block is:

$$p = P(D < 0) = \Phi\left(\frac{0 - \Delta\mu_{\text{eff}}/B}{\sqrt{2\sigma^2/B}}\right) \tag{10}$$

$$= \Phi\left(-\frac{\Delta\mu_{\text{eff}}}{\sigma\sqrt{2B}}\right) \tag{11}$$

$$= \Phi\left(-\Delta\mu_{\text{eff}}\sqrt{\frac{d}{2B}}\right) \tag{12}$$

This yields the signal-to-noise ratio:

$$\text{SNR} = \Delta\mu_{\text{eff}}\sqrt{\frac{d}{2B}}$$

For reliable top-$k$ retrieval with $n$ blocks, we need $p < k/n$, which requires $\text{SNR} > \Phi^{-1}(1 - k/n)$.

## C KEY CONVOLUTION DESIGN

To encourage within-block clustering of semantically related tokens, we apply a short convolution operator to the key representations. This section details the specific design choices.

### C.1 ARCHITECTURE

We use a **depthwise causal 1-D convolution** applied to token-level keys before they are used for routing and attention computation. The key design choices are:

**Depthwise convolution.** The convolution is depthwise-separable with `groups=hidden_size`, meaning each channel (dimension) of the key vector is filtered independently. This reduces parameters while maintaining expressiveness across all dimensions.

**Causal structure.** The convolution is causal (left-padded), ensuring that the representation at position $t$ only depends on positions $\{t - W + 1, \ldots, t\}$ where $W$ is the kernel width. This preserves the autoregressive property required for language modeling.

**Kernel size.** We experiment with kernel widths $W \in \{3, 5\}$, denoted as **kconv3** and **kconv5** respectively in our experiments. These short receptive fields allow local signal diffusion without excessive computational overhead.

**Activation and residual.** Following recent work on efficient linear transformers (Yang et al., 2024), we apply:

- **SiLU activation**: $\sigma(x) = x \cdot \text{sigmoid}(x)$ applied element-wise after convolution
- **Residual connection**: The original key is added back to the convolved output

Formally, for a key vector $\mathbf{k}_t \in \mathbb{R}^d$ at position $t$, the transformed key is:

$$\mathbf{k}'_t = \mathbf{k}_t + \text{SiLU}\left(\sum_{\ell=0}^{W-1} \mathbf{W}_\ell \odot \mathbf{k}_{t-\ell}\right)$$

where $\mathbf{W}_\ell \in \mathbb{R}^d$ are the learnable kernel weights for each lag $\ell$, and $\odot$ denotes element-wise multiplication (due to depthwise structure).

## C.2 Effect on Routing

The key convolution is applied *before* block centroid computation. For block $j$, the centroid used for routing becomes:

$$\tilde{\mathbf{k}}_j = \frac{1}{B} \sum_{\mathbf{k}' \in \mathbf{K}'_j} \mathbf{k}'$$

where $\mathbf{K}'_j$ contains the convolved keys in block $j$.

During training, this design encourages gradient flow between neighboring tokens within a block. When the router learns to select a block containing relevant information, the gradients backpropagate through the convolution, implicitly encouraging nearby tokens to align with the query direction. This increases the number of related tokens $m$ within selected blocks and raises their average affinity $\mu_{\text{cluster}}$, thereby amplifying $\Delta\mu_{\text{eff}}$ according to our statistical model (Section 3).

# D Kernel Implementation Details

## D.1 FlashMoBA Top-K Selection and Index Reformatting

Our algorithm for top-k selection and index reformatting, which we name Flash TopK, is divided into three logical stages, each implemented as a fused kernel to minimize HBM data transfers.

**1. Computation of Block Centroids.** We first compute key-block centroids using a fused Triton kernel (Algorithm 2). This step produces a centroid matrix $\tilde{\mathbf{K}}$ that is $B$ times smaller than the original key matrix $\mathbf{K}$, significantly reducing subsequent HBM accesses.

---

**Algorithm 2** Fused Key-Block Centroid Computation

---

**Require:** Key matrix $\mathbf{K} \in \mathbb{R}^{N \times d}$, block size $B$.
1: Partition $\mathbf{K}$ into blocks $\{\mathbf{K}_j\}_{j=0}^{T_k-1}$, where $T_k = \lceil N/B \rceil$.
2: Initialize centroid matrix $\tilde{\mathbf{K}} \in \mathbb{R}^{T_k \times d}$.
3: **for** $j \in [0, T_k - 1]$ **do in parallel**
4:     $\tilde{\mathbf{K}}[j, :] \leftarrow \frac{1}{|\mathbf{K}_j|} \sum_{\mathbf{v} \in \mathbf{K}_j} \mathbf{v}$                     ▷ Compute mean of each block
5: **end for**

---

**2. Fused Top-K Selection.** With the block centroids pre-computed, a second fused kernel identifies the top-k blocks for each query, adopting the tiling strategy from FlashAttention-2. For each block of queries loaded into SRAM, the kernel iterates through the centroid matrix $\tilde{\mathbf{K}}$ in chunks. It computes gating scores and maintains a running list of the top-k indices and their corresponding scores for each query on-chip. This update is performed with a bubble sort algorithm which is highly efficient for $k \ll N$, as detailed in Algorithm 3. This process avoids materializing the full score matrix to HBM.

**3. Reformatting Indices to Varlen Format.** To facilitate efficient densification in the main attention pass, the query-centric top-k indices must be reformatted into a key-block-centric, variable-length (*varlen*) layout. This layout stores, for each key-block, a compact list of the queries that attend to it. This transformation is implemented as a two-stage epilogue, detailed in Algorithm 4. The first kernel calculates the memory offsets for each key-block via a prefix sum over the histogram computed in the top-k selection stage. The second kernel then reads the query-centric indices and scatters the corresponding query IDs into their correct positions in the final varlen array.

## D.2 FlashMoBA Forward Pass Kernel

The core of our forward pass kernel is the "gather-and-densify" strategy, which allows us to use the efficient dense computation patterns of FlashAttention-2 in a sparse context. To manage this, we distinguish between two types of processing blocks:

---

**Algorithm 3** Fused Top-K Selection

---

**Require:** Matrices $\mathbf{Q} \in \mathbb{R}^{N \times d}$ and $\tilde{\mathbf{K}} \in \mathbb{R}^{\lceil \frac{N}{B} \rceil \times d}$ in HBM, MoBA key-block size $B$, MoBA top-k $k$, block sizes $B_c, B_r$

1: Divide $\mathbf{Q}$ into $T_r = \lceil \frac{N}{B_r} \rceil$ blocks $\mathbf{Q}_i$ of dimensions $B_r \times d$ each, and divide $\tilde{\mathbf{K}}$ into $T_c = \lceil \frac{T_k}{B_c} \rceil$ blocks $\tilde{\mathbf{K}}_j$ of dimensions $B_c \times d$ each.

2: Initialize the top-k indices matrix $\mathbf{I} \in \mathbb{Z}^{N \times k}$ in HBM.

3: **for** $0 \le i < T_r$ **do in parallel**

4:     Load $\mathbf{Q}_i$ from HBM to SRAM.

5:     On chip, initialize $\mathbf{I}_i = (-1)_{B_r \times k}$ and $\mathbf{T}_i = (-\infty)_{B_r \times k}$ in SRAM for the actual attention scores.

6:     **for** $0 \le j < T_c$ **do**

7:         Load $\tilde{\mathbf{K}}_j$ from HBM to on-chip SRAM.

8:         On-chip, compute scores $\mathbf{S} = \mathbf{Q}_i \tilde{\mathbf{K}}_j^\top$ and apply the causal mask.

9:         **for** $0 \le r < B_r$ **do in parallel**

10:            **for** $0 \le c < B_c$ **do**

11:                Update the top-k indices and scores if necessary using bubble sort.

12:            **end for**

13:         **end for**

14:     **end for**

15:     Write the final $\mathbf{I}_i$ to HBM as the $i$-th block of $\mathbf{I}$.

16: **end for**

---

**Algorithm 4** Reformatting Indices to Varlen Key-Block-Major

---

**Require:** Query-centric indices $\mathbf{I} \in \mathbb{Z}^{N \times k}$, key-block counts $\mathbf{C} \in \mathbb{Z}^{T_k}$.

1: Initialize offset array $\text{Offset} \in \mathbb{Z}^{T_k}$ and varlen array $A \in \mathbb{Z}^{(N \times k)}$.

2: $\text{Offset}[j] \leftarrow \sum_{i=0}^{j-1} \mathbf{C}[i]$, for $j = 0, \ldots, T_k - 1$        ▷ Stage 1: Compute offsets

3: Reset temporary counts $\mathbf{C}$ to zero.

4: **for** each query $i$ and its selected block index $b \in \mathbf{I}[i, :]$ **do in parallel** ▷ Stage 2: Scatter indices

5:     $p \leftarrow \text{atomicAdd}(\&\mathbf{C}[b], 1)$

6:     $A[\text{Offset}[b] + p] \leftarrow i$

7: **end for**

---

- **Logical Blocks:** These are the large, contiguous blocks of queries ($\mathbf{Q_i}$) and keys ($\mathbf{K_j}$) that the kernel iterates over in its outer loops. Importantly, a logical key block is equivalent to a MoBA key block.

- **Physical Blocks:** These are the smaller tiles (e.g., $64 \times 64$ or $128 \times 128$) that are loaded into SRAM for the actual matrix multiplication. The optimal size of these blocks depends on the specific GPU architecture and model head dimension.

As described in Algorithm 1, our kernel assigns a logical query block $\mathbf{Q_i}$ to each thread block. This thread block then iterates through all logical key blocks $\mathbf{K_j}$. For each $(\mathbf{Q_i}, \mathbf{K_j})$ pair, it uses the varlen indices to identify the relevant subset of queries within $\mathbf{Q_i}$. This sparse subset of queries is then processed in dense physical blocks: one physical block of queries is gathered from HBM into a dense SRAM buffer for computation.

This two-level blocking strategy is the key to our kernel's performance. By caching a dense physical block of gathered queries in SRAM, it can be reused for matrix multiplication against all the physical tiles that form a logical key block $\mathbf{K_j}$. This reuse effectively amortizes the cost of irregular gather operations over several highly efficient, dense GEMMs. The dimensions of the logical blocks present a tuning opportunity: increasing the logical key block size enhances the query reuse factor, while increasing the logical query block size results in a larger, more computationally efficient subset of gathered queries, albeit at the cost of higher on-chip memory usage.

### D.3 FLASHMoBA BACKWARD PASS KERNEL

The backward pass adapts the memory-efficient design of FlashAttention-2 and is implemented as a sequence of three fused kernels. First, a preprocessing kernel computes the per-query dot product between the output gradients and the outputs ($\mathbf{dO} \cdot \mathbf{O}$), which is required for the softmax gradient calculation. The main kernel then parallelizes the computation across the key dimension, where each thread block is responsible for a logical key block ($\mathbf{K_j}, \mathbf{V_j}$). For its assigned block, a thread block uses the pre-computed varlen indices to gather the corresponding sparse subsets of queries ($\mathbf{Q}$), and output gradients ($\mathbf{dO}$) from HBM. With this data densified on-chip, the kernel recomputes the attention scores to avoid storing the large attention matrix, and then calculates the gradients $\mathbf{dK_j}, \mathbf{dV_j}$, and partial $\mathbf{dQ}$. The gradients for the current block, $\mathbf{dK_j}$ and $\mathbf{dV_j}$, are written directly to HBM, while the partial query gradients are atomically added to a high-precision global buffer, $\mathbf{dQ_{accum}}$. Finally, a post-processing kernel converts the accumulated $\mathbf{dQ_{accum}}$ buffer into the final data type and writes it to the output gradient tensor $\mathbf{dQ}$. Algorithm 5 details this process.

---

**Algorithm 5** FlashMoBA Backward Pass

---

**Require:** Matrices $\mathbf{Q}, \mathbf{K}, \mathbf{V}, \mathbf{O}, \mathbf{dO} \in \mathbb{R}^{N \times d}$ in HBM, vector $L \in \mathbb{R}^N$ in HBM, MoBA arrays
    $C, A$, Offset and block sizes $B_c, B_r$.
1: Divide $\mathbf{K}, \mathbf{V}$ into $T_c = \lceil N/B_c \rceil$ blocks $\mathbf{K_j}$ and $\mathbf{V_j}$, of size $B_c \times d$ each.
2: Initialize $\mathbf{dQ} = (0)_{N \times d}$ in HBM. Divide $\mathbf{dK}, \mathbf{dV} \in \mathbb{R}^{N \times d}$ into $T_c$ blocks $\mathbf{dK_j}$ and $\mathbf{dV_j}$, of
    size $B_c \times d$ each.
3: Compute $D = \text{rowsum}(\mathbf{dO} \circ \mathbf{O}) \in \mathbb{R}^N$ (pointwise multiply), write $D$ to HBM.
4: **for** $0 \le j < T_c$ **do**
5:     Load $\mathbf{K_j}, \mathbf{V_j}$ from HBM to on-chip SRAM.
6:     Initialize $\mathbf{dK_j} = \mathbf{dV_j} = (0)_{B_c \times d}$ on SRAM.
7:     Let $\mathcal{I}_j = \{A_p \mid \text{Offset}_j \le p < \text{Offset}_j + C_j\}$ be the set of $C_j$ query indices that attend to
    block $j$.
8:     The processing of these queries is tiled into $T_r = \lceil C_j/B_r \rceil$ blocks.
9:     **for** $0 \le i < T_r$ **do**
10:        Gather sparse $\mathbf{Q_i^{(j)}}, \mathbf{O_i^{(j)}}, \mathbf{dO_i^{(j)}}, L_i^{(j)}, D_i^{(j)}$ from HBM to SRAM in a dense format.
11:        On chip, compute $\mathbf{S_{ij}} = \mathbf{Q_i^{(j)}} \mathbf{K_j^T} \in \mathbb{R}^{B_r \times B_c}$.
12:        On chip, compute $\mathbf{P_{ij}} = \exp(\mathbf{S_{ij}} - L_i^{(j)}) \in \mathbb{R}^{B_r \times B_c}$.
13:        On chip, compute $\mathbf{dV_j} \leftarrow \mathbf{dV_j} + (\mathbf{P_{ij}})^T \mathbf{dO_i^{(j)}} \in \mathbb{R}^{B_c \times d}$.
14:        On chip, compute $\mathbf{dP_{ij}} = \mathbf{dO_i^{(j)}} \mathbf{V_j^T} \in \mathbb{R}^{B_r \times B_c}$.
15:        On chip, compute $\mathbf{dS_{ij}} = \mathbf{P_{ij}} \circ (\mathbf{dP_{ij}} - D_i^{(j)}) \in \mathbb{R}^{B_r \times B_c}$.
16:        Atomically add $\mathbf{dS_{ij}} \mathbf{K_j}$ to $\mathbf{dQ_{accum_i}^{(j)}}$.
17:        On chip, compute $\mathbf{dK_j} \leftarrow \mathbf{dK_j} + \mathbf{dS_{ij}^T} \mathbf{Q_i^{(j)}} \in \mathbb{R}^{B_c \times d}$.
18:     **end for**
19:     Write $\mathbf{dK_j}, \mathbf{dV_j}$ to HBM.
20: **end for**
21: **return** $\mathbf{dQ}, \mathbf{dK}, \mathbf{dV}$.

---

**Multi-query and grouped-query attention.** Multi-query attention (MQA) Shazeer (2019) and grouped-query attention (GQA) Ainslie et al. (2023) are attention variants that reduce KV cache size during inference by sharing key and value heads across multiple query heads. Like FlashAttention-2, we support these patterns efficiently by remapping attention heads and correctly aggregating over them. Instead of duplicating key/value heads, we adjust indexing to achieve equivalent computation. During backpropagation, gradients for keys and values ($\mathbf{dK}, \mathbf{dV}$) are summed across the shared heads.

