# OpenReview forum: "Optimizing Mixture of Block Attention"
_ICLR.cc/2026/Conference — Submitted to ICLR 2026_

### Official Review · Reviewer_nD1f · 2025-10-16

**Soundness:** 3
**Presentation:** 3
**Contribution:** 3
**Rating:** 6
**Confidence:** 2

**Summary:**

This paper addresses the limitations of Mixture of Block Attention (MoBA). The authors develop a statistical model to derive a key signal-to-noise ratio (SNR) related to head dimension and block size, for better router retrieval accuracy. Based on the SNR formula, the paper uses smaller block sizes and larger head dimensions. Other contributions include key-convolution and efficient implementation.

**Strengths:**

1. Theoretical Framework: The paper introduces a novel signal-to-noise ratio (SNR) model that provides clear and actionable design principles. This provides guidelines for the selection of head dimension and block size.
2. High-Performance CUDA Kernel: FlashMoBA is a well-engineered, hardware-aware CUDA kernel. The Tiled-Topk is especially useful.
3. Strong Benchmark Results: The optimized MoBA models are shown to match or even outperform dense attention on challenging long-context benchmarks like LongBench and RULER.

**Weaknesses:**

1. Limited Generalizability Due to Small Model Scale: All experiments are conducted on a 340M parameter model. This raises significant questions about whether the paper's core findings would scale to the much larger models.
2. Unsubstantiated Link Between SNR and Experiments: The key experiment in Table 4, designed to validate the SNR theory's dependency on head dimension d, fails to control for model size (line 289). It is unclear if the performance improvements in Table 4 are due to the claimed increase in SNR or simply due to the larger model capacity. This methodological flaw means the empirical evidence for the paper's central theoretical claim is not as conclusive as presented.

**Questions:**

There are still rooms in the main text. You should put some algorithm into the main text instead of the appendix, such as the algorithm of Tiled-Topk.

---

> ### Author Response · Authors · 2025-12-03
> **Response to Reviewer nD1f**
>
> Thank you for your suggestions regarding scale and algorithmic clarity. We have incorporated these into the updated manuscript.
>
> 1. Scalability to 1B Parameters:
>
> We have addressed the question of scale by training 1B parameter models on 100B tokens. The results (Tables 2, 4, 6) confirm our findings hold at this scale: MoBA-128 consistently matches or outperforms dense attention. Furthermore, the efficiency gains are even more critical at this scale—our optimized kernel now achieves a 14.7x speedup over FlashAttention-2 at 128K context length.
>
> 2. Isolating the SNR Mechanism:
>
> We have streamlined our experimental validation to focus on Block Size ($B$) as the primary variable. Since $SNR \propto \sqrt{d/B}$, decreasing $B$ increases SNR without adding model parameters. The results in Figure 2 and Table 1 show a clear monotonic improvement as $B$ decreases, providing unconfounded empirical support for the SNR theory.
>
> 3. Algorithms in Main Text:
>
> To improve clarity, we have moved Algorithm 1 (FlashMoBA Forward Pass) to the main text (Page 5), better illustrating the "gather-and-densify" mechanism that enables our speedups.

---

### Official Review · Reviewer_okij · 2025-10-29

**Soundness:** 3
**Presentation:** 2
**Contribution:** 2
**Rating:** 4
**Confidence:** 4

**Summary:**

This paper improves upon the previous sparse attention method, MoBA, through both theoretical analysis and kernel optimization. It analyzes MoBA’s performance from a signal-to-noise ratio (SNR) perspective and identifies that smaller block sizes and larger head dimensions yield performance benefits. To support these findings, the paper implements FlashAttention-style tiling optimizations in the kernel, making MoBA efficient under these settings.

**Strengths:**

1. The paper offers a statistical view that links MoBA hyperparameters to the SNR of attention computation. Although the connection between SNR and end-to-end model performance is not formally derived, the analysis provides a useful proxy for selecting better MoBA hyperparameter configurations.
2. The implementation of a FlashAttention-style MoBA kernel makes the approach practical even with small block sizes. The kernel achieves comparable speed to FlashAttention on short sequences and delivers speedups for long sequence inputs.

**Weaknesses:**

1. Experimental setup: The model architecture setup introduces confounding factors. While the paper claims to focus on optimizing MoBA performance, the model architecture employs sliding window attention (SWA) in half of the layers and involves dense attention in others, limiting the proportion of true MoBA layers. This mixture complicates the attribution of performance improvements and makes it unclear how much gain comes from MoBA, instead of SWA or dense attention components.
2. Key convolution description: The role of the short convolution on keys is not clearly presented. It is unclear how convolution enhances $\Delta_{\mu_{\text{eff}}}$, and additional explanation or intuition is needed. Moreover, implementation details are missing in the main paper. According to Table 1, the performance benefits from adding convolution appear inconsistent.

**Questions:**

1. Dense attention can be viewed as MoBA with block size (1) and top-(K = N). Based on the SNR analysis, smaller block sizes should yield better performance. Why then do most MoBA implementations outperform dense attention across benchmarks in Table 1? Could this discrepancy reflect other dominant factors, such as insufficient training data or incomplete convergence?
2. Could you provide more details on the rationale and implementation of key convolution in the main paper?

---

> ### Author Response · Authors · 2025-12-03
> **Response to Reviewer okij**
>
> Thank you for your constructive feedback. We have refined the manuscript to better articulate the mechanisms driving MoBA's performance.
>
> 1. Why Sparse Outperforms Dense (The "Dilution" Paradox):
>
> You raised an excellent point regarding why MoBA outperforms dense attention. We have added a discussion on "Attention Dilution" in Section 5.2. In dense attention on long sequences, the softmax probability mass is distributed thinly across thousands of irrelevant tokens, drowning out the signal. MoBA’s hard routing acts as a denoising filter, forcing the model to allocate probability mass only to the most relevant blocks. Our experiments at 1B scale (Tables 2 & 6) show MoBA achieving lower perplexity and higher accuracy than dense baselines, confirming this effect.
>
> 2. Key Convolution Details:
>
> We have added a dedicated Appendix C detailing the Key Convolution design. It is a short depthwise causal convolution (kernel size 3 or 5) applied to keys before block centroid calculation. This creates a "smoothing" effect, ensuring that if a single token is relevant, its neighbors also receive high scores. This increases the block's average score ($\mu_{cluster}$ in our SNR model), amplifying the signal for the router.
>
> 3. Experimental Setup (Hybrid Architecture):
>
> We interleave SWA and MoBA to ensure a rigorous comparison with state-of-the-art efficient baselines. Since our Dense baseline also uses this hybrid structure (Dense + SWA layers), the observed performance delta is strictly attributable to the superiority of MoBA layers over Dense layers in this context.

---

### Official Review · Reviewer_yXa6 · 2025-11-04

**Soundness:** 2
**Presentation:** 2
**Contribution:** 3
**Rating:** 4
**Confidence:** 3

**Summary:**

1. The paper presents a statistical analysis of the Mixture of Block attention (MoBA) to motivate the parameters that lead to better performance. Concretely, they propose that the SNR for the MoBA architecture is proportional to the sqrt of the ratio of the head size and the block size: with larger head sizes and smaller block sizes yielding better performance. Additionally, they motivate that semantic clustering in a block also helps improve block retrieval accuracy, leveraging a convolutional layer to demonstrate the point.

2. They provide an efficient implementation for the MoBA architecture in the spirit of FA that computes the top k sparse selection mask without materializing the entire mask, uses it to index into the KV cache to subsequently compute dense attention with the FA methodology , and then finally scatter back the results.

**Strengths:**

1. The paper motivates a better understanding of what aspects contribute to the improvement of the MoBA architecture. The characterization of the SNR as a function of the head size and block size is, to the best of my knowledge, novel and provides a good basic approximation on the framework to tune the performance of MoBA attention.

2. Their kernel implementation is particularly helpful for encouraging the broarder adaptation of the architecture. Given that attention tends to be a bottleneck for a number of new tasks, this is very helpful.

**Weaknesses:**

The following are my concerns with the paper:

1. The core contribution of the paper is relegated to the Appendix. This makes the paper a bit hard to follow, and given that the results motivate the majority of the paper, I do think that at least a part of it should be featured in the main paper.

2. There are a number of assumptions made in the statistical analysis that may not hold true and at the very least merit some grounding with experimental results: L805 makes the assumption that q^Tk are independant dot products, however [1] argue that the dot product are correlated, accounting for a factor of O(d) and not O(sqrt(d)) as the authors propose. Likewise, the authors use the argument for CLT on large B to motivate characterising the delta between the informative and non informative dot products as a normal distribution, however, we subsequently move to argue that B should be small for improved performance. Thus there seems to be a tension between the proposed theory and subsequent proposed improvements.

3. It would be good to have experiments that directly validate the proposed theory. Concretely, for an S-NIAH task, one can compute the estimator for the score difference directly (the block containing the needle vs all other blocks, aggregated across all attention masks). One way to show the validity of the proposed SNR metric would be to plot the estimator as a function of d and B, and then subsequently show that it does follow the proposed trajectory.

4. The models that the authors investigate have a pretty high degree of global attention (50%). This itself can potentially act as a confounder and mask an pitfalls of the proposed algorithm. It would be more informative to have both an analysis of the proposed algorithm scales compared to vanilla dense and vanilla MoBA (i,e comparing to Section 3.1 in [2]) and how the difference in loss changes with introducing additional blocks of MoBA with lower block size / higher attention heads.

5. The experimental results are somewhat counterintuitive: according to the authors' proposal, the performance should keep improving with reduced block sizes. While that is true for Table 1, on Table 3 the trends do not hold. I would have expected the high SNR should be additionally more effective for the longer context evaluations, but that does not seem to be the case ?

6. For the experiments with higher d, the authors keep the number of heads fixed. Thus, the number of parameters (and consequently compute) for the models with higher d is more, since num parameters scales with O(d^2). This makes the ablation experiment with higher d values hard to compare, since they are not IsoFlops, and introduces an additional confounder: do the models with higher d values improve because of more parameters, or because of the better SNR value.

7. The results on the NIAH benchmark seem quite low: for a 32k - 64k context length, this task usually is taken as a necessary but not sufficient task for long context modeling: in fact even the MoBA authors demonstrate a 100% accuracy upto a 1M context window. However, in Table 2, even on the subset of 200 examples, the accuracy seems to be very low for 32k and 64k context lengths. This seems a bit off: it might be because the authors tried zero shot with RoPE for length extrapolation, which is known to have poor performance. It might be better if the authors adapted the vanilla context length extension strategy of training on longer contexts a bit more with RoPE interpolation, and the using the subsequent models for the evaluations. With the current results, it's hard to understand if the proposed method hurts the long context performance or not.

[1] Yang, Greg, et al. "Tensor programs v: Tuning large neural networks via zero-shot hyperparameter transfer." arXiv preprint arXiv:2203.03466 (2022).

[2] Lu, Enzhe, et al. "Moba: Mixture of block attention for long-context llms." arXiv preprint arXiv:2502.13189 (2025).

**Questions:**

For the reduction in the block size experiments, I was not sure of the differences between the proposed experiments by the authors and the experiments on sparsity / granularity tradeoffs of MoBA presented in [1] (section Ablation Study on Fine-Grained Block Segmentation). Would it be possible to clarify the same ?


[2] Lu, Enzhe, et al. "Moba: Mixture of block attention for long-context llms." arXiv preprint arXiv:2502.13189 (2025).

---

> ### Author Response · Authors · 2025-12-03
> **Response to Reviewer yXa6**
>
> We appreciate the reviewer's detailed critique, particularly the recommendation to empirically validate the router's scoring mechanism and the identification of confounding variables in the head dimension analysis. These points have significantly strengthened the manuscript.
>
> 1. Direct Empirical Validation of Router SNR
>
> Regarding the validation of the proposed theory, we implemented a direct probe to measure the estimator for the score difference on an S-NIAH task. We instrumented the router to measure the "Signal-to-Noise" gap using a fixed 32-token needle inserted at token index 2048. We analyzed the Top 10% retrieval heads, as averaging across specialized attention heads dilutes the retrieval signal.
>
> The results provide robust empirical confirmation that smaller blocks amplify the routing signal:
>
> | **Model Config** | **Block Size (B)** | **Signal-Noise Gap (Signal - Max Noise)** | **Improvement Factor** |
> | ---------------- | ------------------ | ----------------------------------------- | ---------------------- |
> | **MoBA-512**     | 512                | 2.78                                      | -                      |
> | **MoBA-256**     | 256                | 6.18                                      | **2.2x**               |
> | **MoBA-128**     | 128                | 13.76                                     | **4.9x (vs 512)**      |
>
> As predicted, the gap increases monotonically as block size decreases. The observed scaling is approximately $\propto 1/B$ (stronger than the conservative $1/\sqrt{B}$ prediction), indicating that smaller blocks not only reduce noise variance but also concentrate the signal to reduce dilution. **This empirically validates the mechanism described in Section 3.**
>
> 2. Removing Confounders (Head Dimension)
>
> The concern regarding parameter count scaling ($O(d^2)$) in the head dimension experiments was well-founded. We have removed the head dimension ablation from the paper. The revised manuscript strictly validates the SNR theory by varying Block Size ($B$) while holding $d$ and total model capacity constant. The results (Figure 2) show that performance improves monotonically as $B$ decreases, effectively isolating the SNR benefit from capacity changes.
>
> 3. Statistical Assumptions
>
> Regarding the independence assumption and CLT usage: While strict independence is a theoretical simplification to derive a tractable heuristic ($d/B$), the probe experiment above demonstrates that the trend holds robustly in practice. Even if correlations exist, the architectural guidance—that minimizing $B$ maximizes retrieval accuracy—remains valid and is now empirically verified.
>
> 4. NIAH Performance and Scale
>
> Regarding the NIAH results:
>
> - **Scale Upgrade:** Experiments have been upgraded from 340M/70B tokens to **1B parameters trained on 100B tokens**.
> - **Performance:** The new 1B MoBA-128 models achieve **100% accuracy** on S-NIAH-1 at 32K context and significantly outperform dense attention at 64K, where dense attention collapses to 0% due to dilution.
> - **Context:** These models are pre-trained on **FineWeb-Edu** without specialized long-context fine-tuning. The fact that MoBA achieves 100% retrieval zero-shot where dense attention fails confirms the effectiveness of the method relative to baselines.
>
> 5. Global Attention and Hybrid Architecture
>
> The hybrid SWA/MoBA architecture is utilized because it has become the standard for efficient long-context modeling (e.g., Command-A, Swan-GPT, GPT-OSS). Since the Dense baseline also employs this hybrid structure (Dense + SWA), the performance delta is strictly attributable to the MoBA layers.
>
> 6. Comparison to Lu et al. (2025)
>
> Regarding the difference from the block segmentation ablation in Lu et al. (2025):
>
> - **Theoretical vs. Empirical:** Lu et al. presented an empirical observation without the theoretical framework (SNR) to explain *why* smaller blocks are superior.
>
> - **Practicality:** Crucially, Lu et al. lacked the hardware implementation to make small blocks feasible; their method becomes slower than dense attention at small $B$. Our contribution includes the **FlashMoBA kernel** (Algorithm 1) that makes $B=128$ practical, achieving a **14.7x speedup** over FlashAttention-2 where the original implementation fails (OOM).

---

### Author Response · Authors · 2025-12-03
**General Response**

We thank the reviewers for their insightful feedback, which has helped us significantly strengthen the empirical validation and experimental rigor of our work. We have submitted an update to the manuscript that directly addresses the core questions regarding model scale, variable control, and the validation of our theoretical model.

**Key Updates in this Revision:**

1. **Scaled up to 1B Parameters:** We extended our experiments from 340M to **1B parameter models**, trained on **100B tokens** (previously 70B). The results demonstrate that MoBA's benefits are scalable and robust: our MoBA-128 models consistently match or outperform dense attention baselines, proving the method's efficacy at larger scales.
2. **Direct Empirical Validation of Router Mechanism:** To rigorously validate our SNR theory, we conducted a new internal probe experiment (detailed in the response to Reviewer yXa6). We measured the "Signal-to-Noise" gap inside the router during inference. The results empirically confirm our theoretical prediction: halving the block size doubles the score gap between the target block and noise blocks, providing concrete evidence for the mechanism driving our performance gains.
3. **Refined Experimental Controls:** We have streamlined our validation strategy to focus strictly on varying Block Size ($B$) while holding model capacity constant. We removed the head dimension ablation experiments to eliminate potential confounding factors, ensuring that the reported gains are strictly attributable to the improved Signal-to-Noise Ratio (SNR).
4. **Algorithmic Transparency:** We moved the core **FlashMoBA Forward Pass algorithm (Algorithm 1)** to the main text and added a dedicated Appendix C detailing the Key Convolution mechanism, significantly improving the paper's self-contained clarity.
5. **Improved Efficiency:** Our optimized kernels now achieve up to **14.7x speedup** over FlashAttention-2 at 128K context length (improved from the previously reported 9x), making the theoretically optimal small-block configurations practically deployable.

---

### Meta-Review · Area_Chair_5rXy · 2025-12-29

**Summary:**

This paper studies mixture of block attention (MoBA), a new variant of the attention mechanism designed to efficiently process long contexts. The paper makes two main contributions. From a theoretical perspective, it introduces a signal-to-noise ratio (SNR) analysis that suggests smaller block sizes lead to higher accuracy. From a practical perspective, the authors propose a new CUDA kernel that enables MoBA with small block sizes. The paper also includes controlled experiments and reports results for two model families pre-trained from scratch.

**Reviewer Concerns:**

Several reviewers have raised concerns about the theoretical result. After reading the rebuttal, this aspect is still confusing to me. First, the theoretical result needs to be made more rigorous. For example, it relies on assumptions that are only stated in the appendix (e.g., assuming the dot products <q,k> are independent). In addition, when the paper states, “We model the router’s task by treating the dot products between a query q and the keys as random variables,” it is important to clearly specify the source of this randomness.

Second, the motivation for the proposed SNR is unclear. The SNR appears to measure the “accuracy” of the router, but ultimately what matters is the output of the attention mechanism, or more importantly the model’s final output. An end-to-end theoretical analysis that connects SNR to the model’s output log-probability’s change would be helpful. The current SNR definition only considers query-key inner products and ignores the role of the values. This can be a significant limitation. For instance, in a hypothetical scenario where all tokens have the same value, the attention output would be identical regardless of which block is selected.

Third, while the proposed SNR depends on both the dimension d and the block size B, all experiments fix d=64. This limits the empirical validation of the theory, as it remains unclear whether \sqrt{d/B​} is the correct scaling.

Reviewers have also raised concerns about the experiments with respect to model size. Although the authors added a larger (1B) model, experiments were only completed for MoBA-128. Moreover, even for the smaller model, results are reported for only three block sizes (128, 256, and 512). Notably, Table 3 in the original submission seems to suggest that MoBA-256 outperforms MoBA-128, which is counterintuitive given the paper’s theoretical claim that smaller block sizes should perform better. While I understand that training such models from scratch is extremely costly, I worry that the paper’s main message, namely that smaller block sizes yield significant quality gains, may be misleading without more comprehensive experimental support.

In summary, while this submission has clear merits (the CUDA kernel, the models (especially if they are open-sourced), and the controlled experimental setup), I believe the paper would benefit a lot from revision and resubmission to address the above-mentioned questions and others raised by the reviewers.

**Reviewer Scores:**

Given the above-mentioned unaddressed issues, I would guess that reviewers keep their scores unchanged.

---

### Decision · Program_Chairs · 2026-01-26

Reject